# Embracing AI: The Imperative Tool for Echo Labs to Stay Ahead of the Curve

**DOI:** 10.3390/diagnostics13193137

**Published:** 2023-10-06

**Authors:** Corina Maria Vasile, Xavier Iriart

**Affiliations:** 1Department of Pediatric and Adult Congenital Cardiology, Bordeaux University Hospital, 33600 Pessac, France; 2IHU Liryc—Electrophysiology and Heart Modelling Institute, Bordeaux University Foundation, 33600 Pessac, France

**Keywords:** artificial intelligence, echo labs, improved efficiency, echocardiography, automated measurements

## Abstract

Advancements in artificial intelligence (AI) have rapidly transformed various sectors, and the field of echocardiography is no exception. AI-driven technologies hold immense potential to revolutionize echo labs’ diagnostic capabilities and improve patient care. This paper explores the importance for echo labs to embrace AI and stay ahead of the curve in harnessing its power. Our manuscript provides an overview of the growing impact of AI on medical imaging, specifically echocardiography. It highlights how AI-driven algorithms can enhance image quality, automate measurements, and accurately diagnose cardiovascular diseases. Additionally, we emphasize the importance of training echo lab professionals in AI implementation to optimize its integration into routine clinical practice. By embracing AI, echo labs can overcome challenges such as workload burden and diagnostic accuracy variability, improving efficiency and patient outcomes. This paper highlights the need for collaboration between echocardiography laboratory experts, AI researchers, and industry stakeholders to drive innovation and establish standardized protocols for implementing AI in echocardiography. In conclusion, this article emphasizes the importance of AI adoption in echocardiography labs, urging practitioners to proactively integrate AI technologies into their workflow and take advantage of their present opportunities. Embracing AI is not just a choice but an imperative for echo labs to maintain their leadership and excel in delivering state-of-the-art cardiac care in the era of advanced medical technologies.

## 1. Introduction

Although artificial intelligence (AI) has a history dating back to the 1950s, recent years have significantly focused on using AI in diagnostic imaging. Machine learning and other AI techniques exhibit a remarkable capacity to discern various patterns within imaging modalities, specifically focusing on echocardiography [1].

AI, by replicating human cognitive functions such as learning and analysis, has extensive applications beyond the medical field, aiding automation and decision-making [2]. In echocardiography, AI can improve data quality, interpretation, and clinical application at various levels, benefiting professionals such as sonographers, echocardiographers, and physicians. Machine learning (ML), a subset of AI, involves algorithms that learn patterns from data to improve performance in specific tasks. ML often requires pre-processing input features to create new variables from data, which can be labor-intensive, especially for complex, high-dimensional datasets such as images and videos. Deep learning (DL), a more specialized ML technique, offers greater adaptability in approximating the underlying data structures, reducing the need for extensive feature engineering to obtain accurate predictions. While DL holds promise for complex datasets such as echocardiograms, it introduces challenges such as complex decision-making (the “black box” effect) and increased computational requirements [3]. Progress in pediatric cardiac imaging has faced complexities due to the complicated nature of pediatric cardiac conditions and their developmental impact. The potential integration of machine learning (ML) in pediatric cardiology, particularly in pediatric echocardiography, is very promising.

Echocardiography, involving dynamic frames, presents unique challenges for AI training compared to static images, like those acquired from computed tomography (CT) or magnetic resonance imaging (MRI) sequences. What sets machine learning models apart is their ability to consider every pixel and its intricate relationships with other pixels, alongside the pertinent clinical metadata. These models can be trained to discern the distinctive features of an echocardiogram. Moreover, this training equips the models to identify images, quantify areas of interest, and establish associations with specific disease patterns [4].

The interplay of clinician interpretation and insights derived from machine learning algorithms enhances echocardiography’s precision. This fusion mitigates inter- and intra-operator variability and brings to light predictive information that may elude human perception. In this context, AI holds the potential to expand the accessibility of clinical expertise and contribute significantly to the field of echocardiography [4].

Echo labs that specialize in echocardiography must recognize the immense benefits that AI integration can bring to their practice. Failure to embrace AI technology and its applications in echocardiography may lead these labs to fall behind their counterparts in the ever-evolving healthcare landscape [5,6].

This paper will explore the importance of integrating artificial intelligence in echo labs and highlight the potential pitfalls that may be faced by labs that choose to neglect this essential technological advancement.

## 2. Methodology

This review aims to provide a perspective on integrating artificial intelligence methods in echo labs. A systematic approach to searching relevant databases, including PubMed, Embase, and Scopus, was meticulously conducted using tailored search terms related to automated measurements, echocardiography, and artificial intelligence in cardiology. Hand-searching journals and meticulous examination of reference lists in seminal articles complemented our approach. To reduce potential selection bias, the two authors diligently assessed each reference’s eligibility. Inclusion criteria included studies and sources published to date in English. This methodology underscores our commitment to providing a comprehensive, informative, and contextually relevant presentation of this difficult topic.

## 3. AI in Echocardiography: Current Landscape

Echocardiography is a fundamental imaging technique essential for accurate diagnosis and optimal therapeutic guidance for people with a spectrum of cardiovascular pathologies. Considered the most important non-invasive cardiac procedure, it has received the strong approval of the European Society of Cardiology. It has become the imaging modality of choice for diagnostic and prognostic evaluation in various cardiac conditions [7,8,9,10,11,12,13].

Rapid progress in developing and integrating AI in healthcare, particularly in the context of cardiac imaging diagnosis, is visible. Specifically, ultrasound equipment manufacturers are at the forefront of AI research and development efforts, complemented by concrete real-world applications, leading to the seamless integration of AI into routine clinical practice. As a result, echocardiographic cardiovascular imaging is undergoing a remarkable transformation, characterized by increasing complexity, while extending its accessibility beyond the domain of cardiologists. Practical applications of AI in echocardiography encompass a range of functionalities, including automation of high-quality image acquisition sequences optimized by computer algorithms, mechanization of measurement processes, and implementation of algorithms designed to interpret cardiac physiological data quickly and accurately. In particular, artificial intelligence algorithms in this context play a key role in enhancing both efficiency and accuracy in echocardiography by decreasing the inherent variability associated with human interpretation [14].

Artificial intelligence has made significant progress in fundamentally revolutionizing echocardiography, with a wide range of applications improving diagnostic capabilities and workflow efficiency. One notable application is image analysis. AI algorithms can automatically recognize echocardiographic sections and detect and classify cardiac structures such as chambers, valves, and myocardium with high accuracy and efficiency.

Artificial intelligence algorithms have greatly advanced image analysis capabilities in echocardiography. Through deep learning and computer vision techniques, these algorithms can automatically analyze echocardiographic images and accurately identify and classify cardiac structures. This includes image optimization and acquisition, segmentation, measurements, global longitudinal strain, disease detection, and periprocedural assessment [14,15,16,17,18,19,20,21,22,23,24,25,26,27,28,29,30]. By automating these tasks, artificial intelligence increases efficiency and minimizes the risk of human error. Currently, echocardiographic image optimization is performed by manual procedures, which inherently depend on the expertise of the cardiac sonographer. However, a shift in perspective is underway as computer algorithms take responsibility for image acquisition and optimization. These algorithms perform image optimization through automatic recognition sequences that adhere to predetermined rules, algorithms, or instructions. Incorporating artificial intelligence algorithms into the image enhancement process offers several notable advantages, including reducing the time required for ultrasound scanning, eliminating image artifacts, mitigating interobserver and interobserver variability, and substantially increasing diagnostic accuracy in cardiac imaging. This transformation represents a key advance in the field with major implications for clinical practice and patient care [16,17].

These algorithms can also segment images, obtaining precise measurements for ventricular volumes, ejection fraction, and different parameter dimensions [14,15,16,17,18,19].

One of the fundamental aspects of echocardiography lies in evaluating and quantifying left ventricular function and size. The assessment of left ventricular function holds significant prognostic value, making it an indispensable component of any echocardiogram report [31]. Numerous techniques are available for measuring left ventricular ejection fraction (LVEF), with the modified Simpson’s biplane method being one of the most commonly employed. This method necessitates manually tracing end-systolic and end-diastolic contours in both apical four- and two-chamber views [26]. However, these manual tracing techniques for biplane disc summation are characterized by considerable variability and exhibit limited correlation with the gold standard, Cardiac Magnetic Resonance (CMR) [1,32].

Presently available AI technology offers the capability for automated echocardiographic measurements. It has been demonstrated that this technology can enhance reproducibility, bridging the gap between experienced and novice readers, while simultaneously improving efficiency and workflow within echocardiography laboratories [33].

In a multicenter study, Knackstedt et al. [25] investigated the feasibility of automated endocardial border detection using a vendor-independent software package. This package employed a machine learning algorithm tailored for image analysis (Auto LV, TomTec-Arena 1.2, TomTec Imaging Systems, Unterschleissheim, Germany). The automated technique proved highly reproducible and exhibited comparability to manual tracing in calculating 2D ejection fraction, left ventricular (LV) volumes, and global longitudinal strain. Importantly, this correlation remained robust when the image quality was good or moderately good, albeit showing a slight reduction in correlation when dealing with poor image quality. Similarly, the results for automated global longitudinal strain demonstrated strong agreement and correlation [25].

The subsequent critical stage in the interpretation of echocardiograms involves the categorization of standardized transthoracic echocardiographic views. In a pioneering effort, Madani and collaborators [18] undertook this task. They leveraged a training dataset comprising 247 real-world echocardiograms, encompassing 200,000 images acquired for clinical purposes, to formulate a unified deep-learning model transcending vendor-specific constraint. This model exhibited remarkable proficiency, achieving a classification accuracy of 98% when tasked with correctly identifying 15 major transthoracic views. Notably, this level of accuracy surpassed that typically achieved by board-certified echocardiographers assigned the same classification mission.

In addition to image analysis, AI algorithms enable automated measurement of various cardiac parameters [19]. They can extract precise measurements of ventricular volumes, wall thickness, ejection fraction, and other important parameters. This automation speeds up the analysis process, improves consistency, and reduces inter-observer variability [14,15,16,17,18,19,20,21,22,23,24,25,26,27,28,29,30].

Automated measurement protocols for 2-dimensional (2D) and 3-dimensional (3D) echocardiographic datasets have become commercially available and are adopted by multiple vendors. These automated measurement packages are crucial in standardizing reproducibility by mitigating human error. The traditional method of determining ejection fractions (EF) in 2D echocardiography, which involves manually tracing endocardial borders, is characterized by its time-intensive nature and reliance on the operator’s expertise.

Moreover, the visual assessment of EF by expert readers is inherently subjective. However, integrating automated border-detection algorithms and identifying end-systolic and end-diastolic frames from the electrocardiogram streamlines the measurement process for cardiac-chamber dimensions, volumes, stroke volume, EF, and wall thickness. Machine learning-assisted, 3D automated assessment of left ventricular (LV) and right ventricular volumes and EF has demonstrated feasibility. This approach offers the advantage of rapid one-minute acquisitions and significantly reduces the need to edit endocardial borders manually.

Two distinct training and validation studies have provided empirical evidence of automatic cine-derived left ventricular ejection fraction (LVEF) accuracy. These studies have shown 90% or higher correlation coefficients when comparing these automated measurements to conventional volume-derived EF determined by clinical readers. This underscores the potential of automated techniques to offer reliable and efficient cardiac assessments in clinical practice [20].

Global longitudinal strain (GLS) denotes the deformation that arises during each myocardial contraction, offering valuable insights into myocardial mechanics through speckle tracking. This technique holds clinical significance for uncovering subclinical ventricular dysfunction that may elude detection by conventional two-dimensional echocardiography. It has gained widespread acceptance, particularly in identifying cardiotoxicity associated with chemotherapy. Furthermore, the distinctive patterns of abnormality observed in GLS can serve as markers for various cardiac pathologies, including cardiac amyloidosis, hypertrophic cardiomyopathy, myocardial infarction, and constriction. Consequently, a burgeoning interest is in leveraging machine learning to evaluate global longitudinal strain [1].

Satle et al. [34] engineered a machine-learning model to assess GLS in a cohort of 200 patients using traditional echocardiographic views, subsequently comparing its performance to standard speckle-tracking software (EchoPac GE). This AI-driven model can autonomously recognize standard apical views, precisely time cardiac events, and measure GLS across a spectrum of cardiac conditions. Impressively, the disparities between the two approaches were minimal, with an absolute difference of merely 1.8%. Furthermore, the AI-based method displayed remarkable efficiency, completing the study in less than 15 s instead of the 5 to 10 min typically required by the conventional approach.

The application of AI in echocardiography includes the fully automated measurement of 2D left ventricular (LV) global longitudinal strain (GLS), a widely adopted practice. This approach enables the rapid assessment of ejection fraction (EF) alongside 2D LV GLS within a mere 8 s, with a high % feasibility rate of 98% and remarkable accuracy. AI achieves this by automatically identifying and categorizing standard views, tracing myocardial motion, and evaluating GLS, particularly in patients with acute myocardial infarction or heart failure [25,34].

AI’s image recognition capabilities are instrumental in disease detection, as they empower the categorization of captured images according to their clinical utility. These capabilities pave the way for disease detection algorithms, especially when integrating Doppler and 2D or 3D measurements. Consequently, this facilitates the analysis of diastolic function, heart failure classification, and the evaluation of valvular lesions, encompassing stenosis or regurgitation.

AI-driven algorithms have identified disease-specific echocardiographic patterns in conditions like hypertrophic cardiomyopathy, amyloidosis, or pulmonary hypertension. Their diagnostic accuracy aligns with that achieved by seasoned clinical readers. Furthermore, there is ongoing development of algorithms geared toward the automatic determination of valvular disease severity [35]. For instance, Moghaddasi et al. [28] employed a machine-learning AI technique to grade mitral valve regurgitation among 139 patients. Impressively, they reported accuracy rates of 99.5% for identifying a normal mitral valve and 99.38%, 99.31%, and 99.59% accuracy for identifying mild, moderate, and severe mitral regurgitation, respectively, with an overall sensitivity of 99.38% and specificity of 99.63%. Additionally, Playford and colleagues [29] conducted a comparative study, evaluating the accuracy of the traditional continuity equation against an AI algorithm designed to identify severe high-gradient aortic stenosis. This AI approach utilized phenotypic characteristics, enhancing the diagnosis of aortic stenosis without referencing the left ventricular outflow tract (LVOT) [27].

Significantly, AI is poised to revolutionize comparing current echocardiograms with previous studies, rendering it both automated and time-efficient. For instance, algorithms can facilitate swift, side-by-side comparisons with analogous images from prior examinations. This streamlined approach substantially reduces the time and effort required for loading and real-time study comparisons. Furthermore, shifting the focus to image comparisons rather than solely relying on prior reports enhances accuracy and aligns with laboratory accreditation requirements [36].

The continuous advancements in AI will also bring about a transformation in the workflow of busy echocardiography laboratories. Currently, unread echocardiograms are prioritized for interpretation based on factors like the patient’s length of stay in the hospital and the presumed urgency of the study, categorized as “stat”, “intensive care unit”, or “routine”. In the future, a pivotal shift will prioritize the most urgent and clinically relevant unread echocardiograms, aligning with evolving clinical demands.

AI’s impact extends to evaluating the suitability of the aortic annulus in patients undergoing transcatheter aortic valve replacement (TAVR). In a single-center study involving 47 patients, AI-driven software was employed to obtain periprocedural aortic annular measurements. These AI-generated measurements were compared with those acquired through traditional 2D transesophageal echocardiography or cardiac computed tomography. The results demonstrated a strong correlation between AI-derived measurements and cardiac computed tomographic data, notably surpassing the performance of transesophageal echocardiographic measurements (correlation coefficient, r = 0.84; *p*-value < 0.0001) [30].

Furthermore, vendor-specific AI protocols are becoming instrumental in evaluating the anatomy of the mitral valve and conducting automated measurements critical for periprocedural mitral clip assessment. These algorithms prioritize precise sizing and offer real-time imaging guidance.

## 4. Benefits and Implications

These current applications of AI in echocardiography have shown promising results and hold immense potential to improve accuracy, efficiency, and personalized care in this field.

Developing and implementing AI algorithms for image analysis, automated measurements, and anomaly detection in echocardiography are promising. They might enhance the accuracy and efficiency of echocardiographic interpretation, reduce the burden on healthcare professionals, and facilitate more effective diagnosis and treatment planning. However, continuous validation, refinement, and collaboration between AI systems and human experts are necessary to ensure their reliability and optimal integration into clinical practice.

One of the main strengths of artificial intelligence in echocardiography is its ability to analyze complex data quickly and precisely [37,38,39]. Significant progress has been achieved in applying AI analysis to echocardiographic images for cart-based equipment and handheld devices. These advancements include automated capabilities to identify specific echocardiography views and segment the heart to precisely quantify parameters like volumes and ejection fraction [40,41,42], thereby allowing more accurate and objective cardiac structure and function assessments, supporting the diagnosis of various cardiovascular diseases.

AI-powered echocardiography systems can also assist in automating repetitive tasks and reduce the workload of healthcare professionals. By automating image acquisition, assessment and analysis, and report generation, AI can streamline workflow and save clinicians time to focus on interpreting results and making clinical decisions. This can lead to faster echocardiography reporting times and improved patient throughput in healthcare facilities [19,43].

Active research in artificial intelligence and precision medicine is moving towards a future where medical professionals and consumers will be equipped with highly personalized diagnostic and therapeutic medical information. The synergy between these two forces has the potential to have a profound impact on the healthcare system, aligning with the ultimate goal of disease prevention and detection at the individual level. This, in turn, can lessen the overall disease burden for the public and reduce preventable healthcare costs.

However, it is important to note that AI in echocardiography is still in its early development and deployment stages. Challenges such as data quality, standardization of imaging protocols, and regulatory considerations need to be addressed for wider adoption and reliable implementation of AI-based systems. Additionally, the ethical implications and potential impact on the clinician–patient relationship should be carefully considered as AI becomes more integrated into clinical practice [17,44].

The power of artificial intelligence in echocardiography has tremendous potential to increase diagnostic accuracy, improve efficiency, and enable personalized patient care. Over time, as AI technology continues to advance and overcome existing challenges, it will likely become an indispensable tool for cardiovascular physicians, ultimately benefiting patients by providing better and more accurate cardiac assessments [45].

The integration of AI into echocardiography offers numerous advantages, including workflow optimization through the automation of tasks such as image acquisition, segmentation, and measurements [17,46,47,48,49,50,51]. This automation enhances efficiency and productivity by reducing the need for manual procedures, ensuring consistent measurement accuracy, and streamlining report generation [17,47,48,49,50,51,52].

Automated measurements, especially for cardiac dimensions and ejection fraction, are essential for accurate diagnoses, significantly reducing manual errors and guaranteeing consistent results [14,52]. This workflow optimization allows healthcare professionals to redirect their expertise towards complex cases and result interpretation, ensuring comprehensive patient evaluations.

Furthermore, AI-driven workflow improvements positively impact patient care by decreasing waiting times for echocardiographic reports. This, in turn, allows healthcare professionals to allocate more time for patient interactions, addressing concerns, and delivering personalized care [53]. AI complements clinical judgment, optimizing resource utilization and enhancing patient experience.

Quality control is pivotal in echocardiography, and AI ensures standardized and accurate reporting across echo labs [54,55]. AI algorithms provide standardized guidelines and measurements, automating the generation of echocardiography reports [54,55]. Additionally, they identify errors and inconsistencies in interpretations, flagging measurement discrepancies and highlighting abnormalities [56,57,58,59]. This ensures thorough and guideline-aligned interpretations, enhancing research outcomes and evidence-based practice.

While AI technology offers significant advantages, it should not replace the expertise of healthcare professionals but rather support and augment clinical judgment. AI’s role is to assist in the diagnostic process, ultimately optimizing resource utilization and ensuring patients receive the highest quality of care.

Another advantage of AI integration is cost efficiency, as it can reduce operational costs and improve financial sustainability. AI technology automates tasks, such as image acquisition and measurement calculations, streamlining workflow processes, and optimizing resource allocation. While initial costs may be associated with AI implementation, the long-term benefits of cost savings and financial sustainability are significant. However, ensuring data security, privacy, and regulatory compliance mitigates potential risks and associated costs. Overall, AI’s contribution to financial sustainability in echo labs is expected to grow as the technology advances and matures, benefiting both patients and healthcare providers [60].

## 5. Potential Challenges and Ethical Considerations

In the evolving landscape of echocardiography, several challenges and ethical considerations need to be addressed. One of the foremost concerns is the imperative need for echo labs to prioritize safeguarding patient data. As AI integration advances within echocardiography, the handling and secure storage of patient data become pivotal considerations. To maintain regulatory compliance and patient trust, echo labs must institute robust security measures. These measures encompass encrypting patient data, ensuring secure storage, and permitting access only to authorized personnel. Strict adherence to privacy regulations, such as the General Data Protection Regulation (GDPR) and the Health Insurance Portability and Accountability Act (HIPAA), remains essential to protect patients’ privacy rights [61,62].

Another crucial aspect is the synergy between healthcare professionals and artificial intelligence systems. While AI algorithms offer advanced analysis and interpretation capabilities, they should be viewed as complementary tools that enhance the clinical judgment of healthcare professionals. Collaboration between AI and human experts is vital to ensure the diagnostic process factors in patient-specific contexts, unique issues, and characteristic elements. Healthcare professionals bring their experience, empathy and critical thinking skills to complement the analytical capabilities of artificial intelligence systems [63].

Ultrasound labs should find a compromise, exploiting the full potential of AI through a collaborative approach while maintaining the central role of healthcare professionals in providing patient-centered care.

## 6. Conclusions

As the medical industry embraces the potential of artificial intelligence, ultrasound labs that cannot integrate this transformative technology will face significant challenges. AI promises increased efficiency, improved accuracy, and personalized echocardiography support. Echocardiography labs that value AI can gain a competitive advantage, benefiting from advanced image analysis, streamlined workflows, and improved diagnostic capabilities. However, it is essential to address ethical considerations and ensure the responsible implementation of AI. Finally, integrating AI into ultrasound labs will be key to providing superior patient care and ensuring a prosperous future in the evolving healthcare landscape.

## Data Availability

More data are available upon request from the corresponding author.

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
