# Peer review of "Embracing AI: The Imperative Tool for Echo Labs to Stay Ahead of the Curve"

_diagnostics, 2023, doi:10.3390/diagnostics13193137_

Round 1
Reviewer 1 Report
The authors attempt to review a very interesting topic of the utility and application of AI in Echocardiography. I commend the authors in providing a thorough literature review of current data on this issue, and for their attempt to organize the data in logical categories that are relevant to an audience that would include echocardiographers, technicians and lab directors.
I found the article very interesting to read though there are some major critiques that need to be addressed:
1) The manuscript could be greatly enhanced by providing some the data/conclusions from the references that are cited, in contrast to just providing the references but with a summary of the findings in those studies. For example, instead of just noting that the AI provides higher accuracy for quantification of LV ejection fraction, the authors could summarize the data i.e "Author et al employed 'proprietary AI software ' in comparing machine learning to expert humans in 'blank #' of echocardiograms that demonstrates improved accuracy compared to 'gold standard measurement'". This is just an example. The same could be applied when talking about the other aspects of AI as well.
2) There is a lot of repetition of the same points regarding the accuracy, automation, efficiency etc of AI in nearly every section of this manuscript. I would suggest shortening the manuscript to limit how repetitive the text is, and the authors may be able to combine the sections given how much overlap and repetition there is.
3) The authors should explain strong statements such as "[AI] leads to ...better patient outcomes in echocardiography". This may not be currently supported by the literature, and if there is data to support this then it should be presented.
Author Response
The authors attempt to review a very interesting topic of the utility and application of AI in Echocardiography. I commend the authors in providing a thorough literature review of current data on this issue, and for their attempt to organize the data in logical categories that are relevant to an audience that would include echocardiographers, technicians and lab directors.
I found the article very interesting to read though there are some major critiques that need to be addressed:
1) The manuscript could be greatly enhanced by providing some the data/conclusions from the references that are cited, in contrast to just providing the references but with a summary of the findings in those studies. For example, instead of just noting that the AI provides higher accuracy for quantification of LV ejection fraction, the authors could summarize the data i.e "Author et al employed 'proprietary AI software ' in comparing machine learning to expert humans in 'blank #' of echocardiograms that demonstrates improved accuracy compared to 'gold standard measurement'". This is just an example. The same could be applied when talking about the other aspects of AI as well.
Thank you for your suggestion. We have revised our manuscript.
2) There is a lot of repetition of the same points regarding the accuracy, automation, efficiency etc of AI in nearly every section of this manuscript. I would suggest shortening the manuscript to limit how repetitive the text is, and the authors may be able to combine the sections given how much overlap and repetition there is.
Our manuscript has been revised.
3) The authors should explain strong statements such as "[AI] leads to ...better patient outcomes in echocardiography". This may not be currently supported by the literature, and if there is data to support this then it should be presented.
The entire manuscript has been revised, and all of your requirements have been addressed.
Reviewer 2 Report
The article is superficial, lacks novelty, and seems to be written by Artificial Intelligence. It is highly repetitive and never goes deep into any of the topics that it describes. There is no quantitative support for the 25 times that mentions efficiency nor for the twelve times that mention accuracy. It seems biased toward the promotion of artificial intelligence. Does not describe any basic definitions between machine learning or deep learning algorithms. The annotations problem for machine learning or the explainability limitations of deep learning are not mentioned. Several sentence lack support and many do not match the references.
Author Response
The article is superficial, lacks novelty, and seems to be written by Artificial Intelligence. It is highly repetitive and never goes deep into any of the topics that it describes. There is no quantitative support for the 25 times that mentions efficiency nor for the twelve times that mention accuracy. It seems biased toward the promotion of artificial intelligence. Does not describe any basic definitions between machine learning or deep learning algorithms. The annotations problem for machine learning or the explainability limitations of deep learning are not mentioned. Several sentence lack support and many do not match the references.
The manuscript has been revised.
Round 2
Reviewer 1 Report
Very enjoyable revised manuscript to read! The information is organized and well presented for sections 1 thru 3, and appreciated the extensive re-writing that was required by the authors.
I do have one major critique: Starting a line 308 thru line 349 in section 3, this information should be under Section 4 "Benefits and Implications" and Section 4, lines 351 thru 529 is very repetitive still and should be synthesized and shortened. I agree that the information is important and useful, but the same points are made over and over again. The first half of the manuscript is really well written, so would apply this to the second half of the manuscript as well. Will then be an excellent read!
Author Response
R1:
Very enjoyable revised manuscript to read! The information is organized and well presented for sections 1 thru 3, and appreciated the extensive re-writing that was required by the authors.
I do have one major critique: Starting a line 308 thru line 349 in section 3, this information should be under Section 4 "Benefits and Implications" and Section 4, lines 351 thru 529 is very repetitive still and should be synthesized and shortened. I agree that the information is important and useful, but the same points are made over and over again. The first half of the manuscript is really well written, so would apply this to the second half of the manuscript as well. Will then be an excellent read!
We appreciate you took the time to review our manuscript.
Lines 308-349 have been moved to Section 4.
Lines 351-529 have been rephrased and shortened.
Reviewer 2 Report
This perspective article partially improved. The current version is more precise, but the article remains redundant. Bias seems to be reduced.
Mayor comments
At least a 30% reduction in length should be implemented before acceptance.
Be careful with words like reasoning (line 36) when referring to artificial intelligence.
Some sentences are too long; please shorten them for clarity.
Speckels and pixels are not the same and the artificial intelligence approach should be clarified.
Minor
Abbreviations mentioned once, like MRI and CMR, or twice, like LVEF, should be removed.
"artificial intelligence (AI)" appears four times in the manuscript, correct, please
Author Response
R2:
This perspective article partially improved. The current version is more precise, but the article remains redundant. Bias seems to be reduced.
Mayor comments
At least a 30% reduction in length should be implemented before acceptance.
If the Editor agrees with this recommendation, we could reduce the number of words. Considering the Journal of Clinical Medicine guidelines, the review manuscript should have at least 4000 words and if we proceed with this recommendation, our word count will not be within the limits.
Be careful with words like reasoning (line 36) when referring to artificial intelligence.
We have modified it accordingly.
Some sentences are too long; please shorten them for clarity.
We have shortened the long sentences.
Speckels and pixels are not the same and the artificial intelligence approach should be clarified.
We do not understand what exactly the reviewer wants.
Minor
Abbreviations mentioned once, like MRI and CMR, or twice, like LVEF, should be removed.
"artificial intelligence (AI)" appears four times in the manuscript, correct, please
We decided to keep even the abbreviations mentioned once.
Round 3
Reviewer 2 Report
For a perspective article, the content is suitable despite that it remains vague in the descriptions. Bias has been reduced. The current version could be acceptable if the editor decides to continue with the publication process.